# LOCAL SEARCH ALGORITHMS FOR RANK-CONSTRAINED CONVEX OPTIMIZATION

**Kyriakos Axiotis**
MIT
kaxiotis@mit.edu

**Maxim Sviridenko**
Yahoo! Research NYC
sviri@verizonmedia.com

## ABSTRACT

We propose greedy and local search algorithms for rank-constrained convex optimization, namely solving $\min_{\text{rank}(A) \leq r^*} R(A)$ given a convex function $R : \mathbb{R}^{m \times n} \to \mathbb{R}$ and a parameter $r^*$. These algorithms consist of repeating two steps: (a) adding a new rank-1 matrix to $A$ and (b) enforcing the rank constraint on $A$. We refine and improve the theoretical analysis of Shalev-Shwartz et al. (2011), and show that if the *rank-restricted* condition number of $R$ is $\kappa$, a solution $A$ with rank $O(r^* \cdot \min\{\kappa \log \frac{R(\mathbf{0}) - R(A^*)}{\epsilon}, \kappa^2\})$ and $R(A) \leq R(A^*) + \epsilon$ can be recovered, where $A^*$ is the optimal solution. This significantly generalizes associated results on sparse convex optimization, as well as rank-constrained convex optimization for smooth functions. We then introduce new practical variants of these algorithms that have superior runtime and recover better solutions in practice. We demonstrate the versatility of these methods on a wide range of applications involving matrix completion and robust principal component analysis.

## 1 INTRODUCTION

Given a real-valued convex function $R : \mathbb{R}^{m \times n} \to \mathbb{R}$ on real matrices and a parameter $r^* \in \mathbb{N}$, the *rank-constrained convex optimization* problem consists of finding a matrix $A \in \mathbb{R}^{m \times n}$ that minimizes $R(A)$ among all matrices of rank at most $r^*$:

$$\min_{\text{rank}(A) \leq r^*} R(A) \tag{1}$$

Even though $R$ is convex, the rank constraint makes this problem non-convex. Furthermore, it is known that this problem is NP-hard and even hard to approximate (Natarajan (1995); Foster et al. (2015)).

In this work, we propose efficient greedy and local search algorithms for this problem. Our contribution is twofold:

1. We provide theoretical analyses that bound the rank and objective value of the solutions returned by the two algorithms in terms of the *rank-restricted condition number*, which is the natural generalization of the condition number for low-rank subspaces. The results are significantly stronger than previous known bounds for this problem.

2. We experimentally demonstrate that, after careful performance adjustments, the proposed general-purpose greedy and local search algorithms have superior performance to other methods, even for some of those that are tailored to a particular problem. Thus, these algorithms can be considered as a general tool for rank-constrained convex optimization and a viable alternative to methods that use convex relaxations or alternating minimization.

**The rank-restricted condition number** Similarly to the work in sparse convex optimization, a restricted condition number quantity has been introduced as a reasonable assumption on $R$. If we let $\rho_r^+$ be the maximum smoothness bound and $\rho_r^-$ be the minimum strong convexity bound only along rank-$r$ directions of $R$ (these are called rank-restricted smoothness and strong convexity respectively), the rank-restricted condition number is defined as $\kappa_r = \frac{\rho_r^+}{\rho_r^-}$. If this quantity is bounded,

one can efficiently find a solution $A$ with $R(A) \leq R(A^*) + \epsilon$ and rank $r = O(r^* \cdot \kappa_{r+r^*} \frac{R(\mathbf{0})}{\epsilon})$ using a greedy algorithm (Shalev-Shwartz et al. (2011)). However, this is not an ideal bound since the rank scales linearly with $\frac{R(\mathbf{0})}{\epsilon}$, which can be particularly high in practice. Inspired by the analogous literature on sparse convex optimization by Natarajan (1995); Shalev-Shwartz et al. (2010); Zhang (2011); Jain et al. (2014) and more recently Axiotis & Sviridenko (2020), one would hope to achieve a logarithmic dependence or no dependence at all on $\frac{R(\mathbf{0})}{\epsilon}$. In this paper we achieve this goal by providing an improved analysis showing that the greedy algorithm of Shalev-Shwartz et al. (2011) in fact returns a matrix of rank of $r = O(r^* \cdot \kappa_{r+r^*} \log \frac{R(\mathbf{0})}{\epsilon})$. We also provide a new local search algorithm together with an analysis guaranteeing a rank of $r = O(r^* \cdot \kappa_{r+r^*}^2)$. Apart from significantly improving upon previous work on rank-restricted convex optimization, these results directly generalize a lot of work in sparse convex optimization, e.g. Natarajan (1995); Shalev-Shwartz et al. (2010); Jain et al. (2014). Our algorithms and theorem statements can be found in Section 2.

**Runtime improvements**   Even though the rank bound guaranteed by our theoretical analyses is adequate, the algorithm runtimes leave much to be desired. In particular, both the greedy algorithm of Shalev-Shwartz et al. (2011) and our local search algorithm have to solve an optimization problem in each iteration in order to find the best possible linear combination of features added so far. Even for the case that $R(A) = \frac{1}{2} \sum_{(i,j) \in \Omega} (M - A)_{ij}^2$, this requires solving a least squares problem on $|\Omega|$ examples and $r^2$ variables. For practical implementations of these algorithms, we circumvent this issue by solving a related optimization problem that is usually much smaller. This instead requires solving $n$ least squares problems with *total* number of examples $|\Omega|$, each on $r$ variables. This not only reduces the size of the problem by a factor of $r$, but also allows for a straightforward distributed implementation. Interestingly, our theoretical analyses still hold for these variants. We propose an additional heuristic that reduces the runtime even more drastically, which is to only run a few (less than 10) iterations of the algorithm used for solving the inner optimization problem. Experimental results show that this modification not only does not significantly worsen results, but for machine learning applications also acts as a regularization method that can dramatically improve generalization. These matters, as well as additional improvements for making the local search algorithm more practical, are addressed in Section 2.3.

**Roadmap**   In Section 2, we provide the descriptions and theoretical results for the algorithms used, along with several modifications to boost performance. In Section 3, we evaluate the proposed greedy and local search algorithms on optimization problems like robust PCA. Then, in Section 4 we evaluate their generalization performance in machine learning problems like matrix completion.

## 2   ALGORITHMS & THEORETICAL GUARANTEES

In Sections 2.1 and 2.2 we state and provide theoretical performance guarantees for the basic greedy and local search algorithms respectively. Then in Section 2.3 we state the algorithmic adjustments that we propose in order to make the algorithms efficient in terms of runtime and generalization performance. A discussion regarding the tightness of the theoretical analysis is deferred to Appendix A.4.

When the dimension is clear from context, we will denote the all-ones vector by $\mathbf{1}$, and the vector that is 0 everywhere and 1 at position $i$ by $\mathbf{1}_i$. Given a matrix $A$, we denote by $\text{im}(A)$ its column span. One notion that we will find useful is that of *singular value thresholding*. More specifically, given a rank-$k$ matrix $A \in \mathbb{R}^{m \times n}$ with SVD $\sum_{i=1}^{k} \sigma_i u^i v^{i\top}$ such that $\sigma_1 \geq \cdots \geq \sigma_k$, as well as an integer parameter $r \geq 1$, we define $H_r(A) = \sum_{i=1}^{r} \sigma_i u^i v^{i\top}$ to be the operator that truncates to the $r$ highest singular values of $A$.

### 2.1   GREEDY

Algorithm 1 (Greedy) was first introduced in Shalev-Shwartz et al. (2011) as the GECO algorithm. It works by iteratively adding a rank-1 matrix to the current solution. This matrix is chosen as the

rank-1 matrix that best approximates the gradient, i.e. the pair of singular vectors corresponding to the maximum singular value of the gradient. In each iteration, an additional procedure is run to optimize the combination of previously chosen singular vectors.

In Shalev-Shwartz et al. (2011) guarantee on the rank of the solution returned by the algorithm is $r^* \kappa_{r+r^*} \frac{R(\mathbf{0})}{\epsilon}$. The main bottleneck in order to improve on the $\frac{R(\mathbf{0})}{\epsilon}$ factor is the fact that the analysis is done in terms of the squared nuclear norm of the optimal solution. As the worst-case discrepancy between the squared nuclear norm and the rank is $R(\mathbf{0})/\epsilon$, their bounds inherit this factor. Our analysis works directly with the rank, in the spirit of sparse optimization results (e.g. Shalev-Shwartz et al. (2011); Jain et al. (2014); Axiotis & Sviridenko (2020)). A challenge compared to these works is the need for a suitable notion of "intersection" between two sets of vectors. The main technical contribution of this work is to show that the orthogonal projection of one set of vectors into the span of the other is such a notion, and, based on this, to define a decomposition of the optimal solution that is used in the analysis.

---

**Algorithm 1** Greedy

---

1: **procedure** GREEDY($r \in \mathbb{N}$ : target rank)
2:     function to be minimized $R : \mathbb{R}^{m \times n} \to \mathbb{R}$
3:     $U \in \mathbb{R}^{m \times 0}$                                                     ▷ Initially rank is zero
4:     $V \in \mathbb{R}^{n \times 0}$
5:     **for** $t = 0 \ldots r - 1$ **do**
6:         $\sigma u v^\top \leftarrow H_1(\nabla R(UV^\top))$     ▷ Max singular value $\sigma$ and corresp. singular vectors $u, v$
7:         $U \leftarrow (U \quad u)$                                  ▷ Append new vectors as columns
8:         $V \leftarrow (V \quad v)$
9:         $U, V \leftarrow \text{OPTIMIZE}(U, V)$
10:     **return** $UV^\top$
11: **procedure** OPTIMIZE($U \in \mathbb{R}^{m \times r}, V \in \mathbb{R}^{n \times r}$)
12:     $X \leftarrow \underset{X \in \mathbb{R}^{r \times r}}{\arg \min} R(UXV^\top)$
13:     **return** $UX, V$

---

**Theorem 2.1** (Algorithm 1 (greedy) analysis). *Let $A^*$ be any fixed optimal solution of (1) for some function $R$ and rank bound $r^*$, and let $\epsilon > 0$ be an error parameter. For any integer $r \geq 2r^* \cdot \kappa_{r+r^*} \log \frac{R(\mathbf{0}) - R(A^*)}{\epsilon}$, if we let $A = \text{GREEDY}(r)$ be the solution returned by Algorithm 1, then $R(A) \leq R(A^*) + \epsilon$. The number of iterations is $r$.*

The proof of Theorem 2.1 can be found in Appendix A.2.

## 2.2 LOCAL SEARCH

One drawback of Algorithm 1 is that it increases the rank in each iteration. Algorithm 2 is a modification of Algorithm 1, in which the rank is truncated in each iteration. The advantage of Algorithm 2 compared to Algorithm 1 is that it is able to make progress without increasing the rank of A, while Algorithm 1 necessarily increases the rank in each iteration. More specifically, because of the greedy nature of Algorithm 1, some rank-1 components that have been added to A might become obsolete or have reduced benefit after a number of iterations. Algorithm 2 is able to identify such candidates and remove them, thus allowing it to continue making progress.

**Theorem 2.2** (Algorithm 2 (local search) analysis). *Let $A^*$ be any fixed optimal solution of (1) for some function $R$ and rank bound $r^*$, and let $\epsilon > 0$ be an error parameter. For any integer $r \geq r^* \cdot (1 + 8\kappa_{r+r^*}^2)$, if we let $A = \text{LOCAL\_SEARCH}(r)$ be the solution returned by Algorithm 2, then $R(A) \leq R(A^*) + \epsilon$. The number of iterations is $O\left(r^* \kappa_{r+r^*} \log \frac{R(\mathbf{0}) - R(A^*)}{\epsilon}\right)$.*

The proof of Theorem 2.2 can be found in Appendix A.3.

---

**Algorithm 2** Local Search

1: **procedure** LOCAL_SEARCH($r \in \mathbb{N}$ : target rank)
2:     function to be minimized $R : \mathbb{R}^{m \times n} \to \mathbb{R}$
3:         $U \leftarrow \mathbf{0}_{m \times r}$                                          ▷ Initialize with all-zero solution
4:         $V \leftarrow \mathbf{0}_{n \times r}$
5:     **for** $t = 0 \ldots L - 1$ **do**                                        ▷ Run for $L$ iterations
6:         $\sigma u v^\top \leftarrow H_1(\nabla R(UV^\top))$       ▷ Max singular value $\sigma$ and corresp. singular vectors $u, v$
7:         $U, V \leftarrow \text{TRUNCATE}(U, V)$                        ▷ Reduce rank of $UV^\top$ by one
8:         $U \leftarrow (U \quad u)$                                          ▷ Append new vectors as columns
9:         $V \leftarrow (V \quad v)$
10:         $U, V \leftarrow \text{OPTIMIZE}(U, V)$
11:     **return** $UV^\top$
12: **procedure** TRUNCATE($U \in \mathbb{R}^{m \times r}, V \in \mathbb{R}^{n \times r}$)
13:     $U \Sigma V^\top \leftarrow \text{SVD}(H_{r-1}(UV^\top))$                ▷ Keep all but minimum singular value
14:     **return** $U\Sigma, V$

---

## 2.3 ALGORITHMIC ADJUSTMENTS

**Inner optimization problem**     The inner optimization problem that is used in both greedy and local search is:

$$\min_{X \in \mathbb{R}^{r \times r}} R(UXV^\top). \tag{2}$$

It essentially finds the choice of matrices $U'$ and $V'$, with columns in the column span of $U$ and $V$ respectively, that minimizes $R(U'V'^\top)$. We, however, consider the following problem instead:

$$\min_{V \in \mathbb{R}^{n \times r}} R(UV^\top). \tag{3}$$

Note that the solution recovered from (3) will never have worse objective value than the one recovered from (2), and that nothing in the analysis of the algorithms breaks. Importantly, (3) can usually be solved much more efficiently than (2). As an example, consider the following objective that appears in matrix completion: $R(A) = \frac{1}{2} \sum_{(i,j) \in \Omega} (M - A)_{ij}^2$ for some $\Omega \subseteq [m] \times [n]$. If we let $\Pi_\Omega(\cdot)$ be an operator that zeroes out all positions in the matrix that are not in $\Omega$, we have $\nabla R(A) = -\Pi_\Omega(M - A)$. The optimality condition of (2) now is $U^\top \Pi_\Omega(M - UXV^\top)V = \mathbf{0}$ and that of (3) is $U^\top \Pi_\Omega(M - UV^\top) = \mathbf{0}$. The former corresponds to a least squares linear regression problem with $|\Omega|$ examples and $r^2$ variables, while the latter can be decomposed into $n$ independent systems $U^\top \left( \sum_{i:(i,j) \in \Omega} \mathbf{1}_i \mathbf{1}_i^\top \right) UV^j = U^\top \Pi_\Omega (M\mathbf{1}_j)$, where the variable is $V^j$ which is the $j$-th column of $V$. The $j$-th of these systems corresponds to a least squares linear regression problem with $|\{i : (i,j) \in \Omega\}|$ examples and $r$ variables. Note that the total number of examples in all systems is $\sum_{j \in [n]} |\{i : (i,j) \in \Omega\}| = |\Omega|$. The choice of $V$ here as the variable to be optimized is arbitrary. In particular, as can be seen in Algorithm 3, in practice we alternate between optimizing $U$ and $V$ in each iteration. It is worthy of mention that the OPTIMIZE_FAST procedure is basically the same as one step of the popular alternating minimization procedure for solving low-rank problems. As a matter of fact, when our proposed algorithms are viewed from this lens, they can be seen as alternating minimization interleaved with rank-1 insertion and/or removal steps.

**Singular value decomposition**     As modern methods for computing the top entries of a singular value decomposition scale very well even for large sparse matrices (Martinsson et al. (2011); Szlam et al. (2014); Tulloch (2014)), the "insertion" step of greedy and local search, in which the top entry of the SVD of the gradient is determined, is quite fast in practice. However, these methods are not suited for computing the *smallest* singular values and corresponding singular vectors, a step required for the local search algorithm that we propose. Therefore, in our practical implementations we opt to perfom the alternative step of directly removing one pair of vectors from the representation $UV^\top$. A simple approach is to go over all $r$ possible removals and pick the one that increases the

---

**Algorithm 3** Fast inner Optimization

---

1: **procedure** OPTIMIZE_FAST($U \in \mathbb{R}^{m \times r}, V \in \mathbb{R}^{n \times r}, t \in \mathbb{N}$ : iteration index of algorithm)
2:      **if** $t \mod 2 = 0$ **then**
3:          $X \leftarrow \underset{X \in \mathbb{R}^{m \times r}}{\arg \min} R(XV^{\top})$
4:          **return**   $X, V$
5:      **else**
6:          $X \leftarrow \underset{X \in \mathbb{R}^{n \times r}}{\arg \min} R(UX^{\top})$
7:          **return**   $U, X$

---

objective by the least amount. A variation of this approach has been used by Shalev-Shwartz et al. (2011). However, a much faster approach is to just pick the pair of vectors $U\mathbf{1}_i, V\mathbf{1}_i$ that minimizes $\|U\mathbf{1}_i\|_2\|V\mathbf{1}_i\|_2$. This is the approach that we use, as can be seen in Algorithm 4.

---

**Algorithm 4** Fast rank reduction

---

1: **procedure** TRUNCATE_FAST($U \in \mathbb{R}^{m \times r}, V \in \mathbb{R}^{n \times r}$)
2:      $i \leftarrow \underset{i \in [r]}{\arg \min} \|U\mathbf{1}_i\|_2\|V\mathbf{1}_i\|_2$
3:      **return**   $\left(U_{[m],[1,i-1]} \quad U_{[m],[i+1,r]}\right), \left(V_{[n],[1,i-1]} \quad V_{[n],[i+1,r]}\right)$      ▷ Remove column $i$

---

After the previous discussion, we are ready to state the fast versions of Algorithm 1 and Algorithm 2 that we use for our experiments. These are Algorithm 2.3 and Algorithm 5. Notice that we initialize Algorithm 5 with the solution of Algorithm 2.3 and we run it until the value of $R(\cdot)$ stops decreasing rather than for a fixed number of iterations.

**Algorithm 2.3** (Fast Greedy). *The Fast Greedy algorithm is defined identically as Algorithm 1, with the only difference that it uses the* OPTIMIZE_FAST *routine as opposed to the* OPTIMIZE *routine.*

---

**Algorithm 5** Fast Local Search

---

1: **procedure** FAST_LOCAL_SEARCH($r \in \mathbb{N}$ : target rank)
2:      function to be minimized $R : \mathbb{R}^{m \times n} \to \mathbb{R}$
3:      $U, V \leftarrow$ solution returned by FAST_GREEDY($r$)
4:      **do**
5:          $U_{\mathrm{prev}}, V_{\mathrm{prev}} \leftarrow U, V$
6:          $\sigma uv^{\top} \leftarrow H_1(\nabla R(UV^{\top}))$      ▷ Max singular value $\sigma$ and corresp. singular vectors $u, v$
7:          $U, V \leftarrow$ TRUNCATE_FAST($U, V$)      ▷ Reduce rank of $UV^{\top}$ by one
8:          $U \leftarrow (U \quad u)$      ▷ Append new vectors as columns
9:          $V \leftarrow (V \quad v)$
10:        $U, V \leftarrow$ OPTIMIZE_FAST($U, V, t$)
11:     **while** $R(UV^{\top}) < R(U_{\mathrm{prev}}V_{\mathrm{prev}}^{\top})$
12:      **return**   $U_{\mathrm{prev}}V_{\mathrm{prev}}^{\top}$

---

## 3    OPTIMIZATION APPLICATIONS

An immediate application of the above algorithms is in the problem of *low rank matrix recovery*. Given any convex distance measure between matrices $d : \mathbb{R}_{m \times n} \times \mathbb{R}_{m \times n} \to \mathbb{R}_{\geq 0}$, the goal is to find a low-rank matrix $A$ that matches a target matrix $M$ as well as possible in terms of $d$: $\underset{\mathrm{rank}(A) \leq r^*}{\min} d(M, A)$ This problem captures a lot of different applications, some of which we go over in the following sections.

## 3.1 LOW-RANK APPROXIMATION ON OBSERVED SET

A particular case of interest is when $d(M, A)$ is the Frobenius norm of $M - A$, but only applied to entries belonging to some set $\Omega$. In other words, $d(M, A) = \frac{1}{2}\|\Pi_\Omega(M - A)\|_F^2$. We have compared our Fast Greedy and Fast Local Search algorithms with the SoftImpute algorithm of Mazumder et al. (2010) as implemented by Rubinsteyn & Feldman (2016), on the same experiments as in Mazumder et al. (2010). We have solved the inner optimization problem required by our algorithms by the LSQR algorithm Paige & Saunders (1982). More specifically, $M = UV^\top + \eta \in \mathbb{R}^{100 \times 100}$, where $\eta$ is some noise vector. We let every entry of $U, V, \eta$ be i.i.d. normal with mean 0 and the entries of $\Omega$ are chosen i.i.d. uniformly at random over the set $[100] \times [100]$. The experiments have three parameters: The true rank $r^*$ (of $UV^\top$), the percentage of observed entries $p = |\Omega|/10^4$, and the signal-to-noise ratio SNR. We measure the normalized MSE, i.e. $\|\Pi_\Omega(M - A)\|_F^2 / \|\Pi_\Omega(M)\|_F^2$. The results can be seen in Figure 1, where it is illustrated that Fast Local Search sometimes returns significantly more accurate and lower-rank solutions than Fast Greedy, and Fast Greedy generally returns significantly more accurate and lower-rank solutions than SoftImpute.

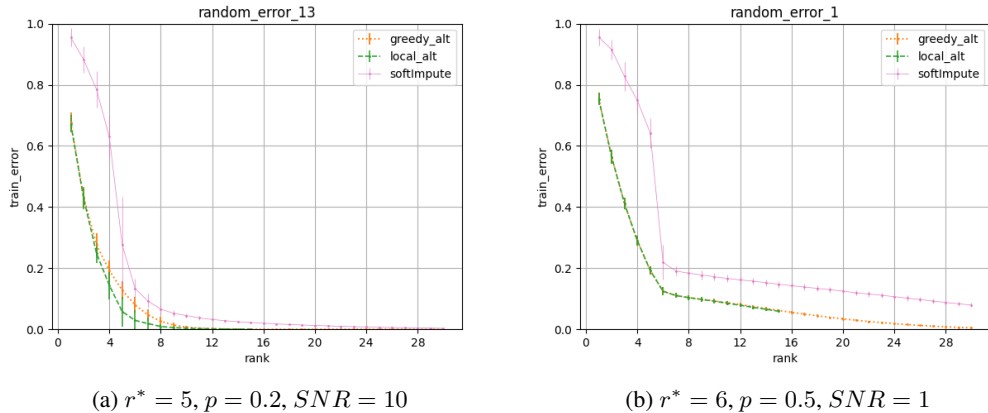

(a) $r^* = 5, p = 0.2, SNR = 10$        (b) $r^* = 6, p = 0.5, SNR = 1$

Figure 1: Objective value error vs rank in the problem of Section 3.1.

## 3.2 ROBUST PRINCIPAL COMPONENT ANALYSIS (RPCA)

The robust PCA paradigm asks one to decompose a given matrix $M$ as $L + S$, where $L$ is a low-rank matrix and $S$ is a sparse matrix. This is useful for applications with outliers where directly computing the principal components of $M$ is significantly affected by them. For a comprehensive survey on Robust PCA survey one can look at Bouwmans et al. (2018). The following optimization problem encodes the above-stated requirements:

$$\min_{\text{rank}(L) \leq r^*} \|M - L\|_0 \tag{4}$$

where $\|X\|_0$ is the sparsity (i.e. number of non-zeros) of $X$. As neither the rank constraint or the $\ell_0$ function are convex, Candès et al. (2011) replaced them by their usual convex relaxations, i.e. the nuclear norm $\| \cdot \|_*$ and $\ell_1$ norm respectively. However, we opt to only relax the $\ell_0$ function but not the rank constraint, leaving us with the problem:

$$\min_{\text{rank}(L) \leq r^*} \|M - L\|_1 \tag{5}$$

In order to make the objective differentiable and thus more well-behaved, we further replace the $\ell_1$ norm by the Huber loss $H_\delta(x) = \begin{cases} x^2/2 & \text{if } |x| \leq \delta \\ \delta|x| - \delta^2/2 & \text{otherwise} \end{cases}$, thus getting: $\min_{\text{rank}(L) \leq r^*} \sum_{ij} H_\delta(M - L)_{ij}$. This is a problem on which we can directly apply our algorithms. We solve the inner optimization problem by applying 10 L-BFGS iterations.

In Figure 2 one can see an example of foreground-background separation from video using robust PCA. The video is from the BMC 2012 dataset Vacavant et al. (2012). In this problem, the low-rank part corresponds to the background and the sparse part to the foreground. We compare three

algorithms: Our Fast Greedy algorithm, standard PCA with 1 component (the choice of 1 was picked to get the best outcome), and the standard Principal Component Pursuit (PCP) algorithm (Candès et al. (2011)), as implemented in Lin et al. (2010), where we tuned the regularization parameter $\lambda$ to achieve the best result. We find that Fast Greedy has the best performance out of the three algorithms in this sample task.

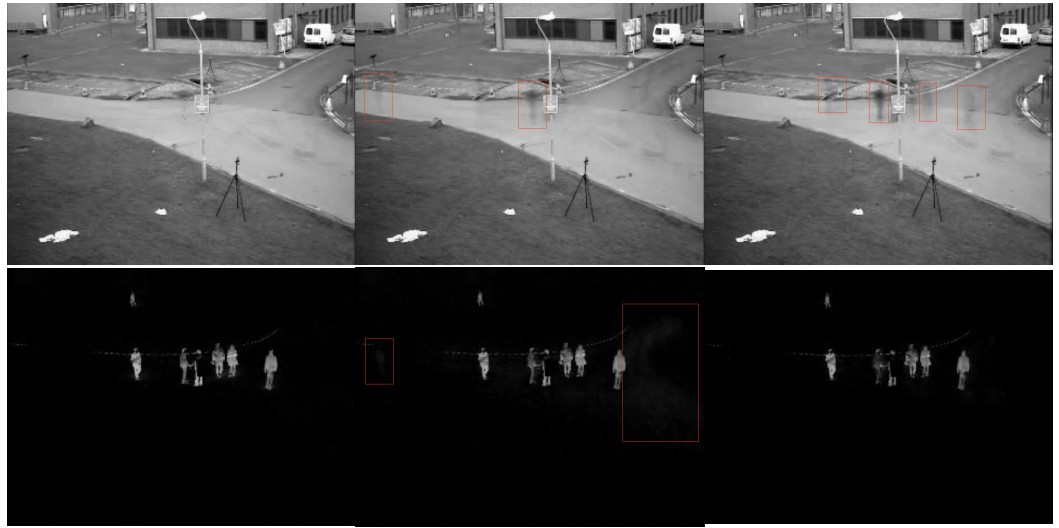

Figure 2: Foreground-background separation from video. From left to right: Fast Greedy with rank=3 and Huber loss with $\delta = 20$. Standard PCA with rank=1. Principal Component Pursuit (PCP) with $\lambda = 0.002$. Both PCA and PCP have visible "shadows" in the foreground that appear as "smudges" in the background. These are less obvious in a still frame but more apparent in a video.

## 4   MACHINE LEARNING APPLICATIONS

### 4.1   REGULARIZATION TECHNIQUES

In the previous section we showed that our proposed algorithms bring down different optimization objectives aggressively. However, in applications where the goal is to obtain a low generalization error, regularization is needed. We considered two different kinds of regularization. The first method is to run the inner optimization algorithm for less iterations, usually 2-3. Usually this is straightforward since an iterative method is used. For example, in the case $R(A) = \frac{1}{2}\|\Pi_\Omega(M-A)\|_F^2$ the inner optimization is a least squares linear regression problem that we solve using the LSQR algorithm. The second one is to add an $\ell_2$ regularizer to the objective function. However, this option did not provide a substantial performance boost in our experiments, and so we have not implemented it.

### 4.2   MATRIX COMPLETION WITH RANDOM NOISE

In this section we evaluate our algorithms on the task of recovering a low rank matrix $UV^\top$ after observing $\Pi_\Omega(UV^\top + \eta)$, i.e. a fraction of its entries with added noise. As in Section 3.1, we use the setting of Mazumder et al. (2010) and compare with the SoftImpute method. The evaluation metric is the normalized MSE, defined as $(\sum_{(i,j)\notin\Omega}(UV^\top - A)_{ij}^2)/(\sum_{(i,j)\notin\Omega}(UV^\top)_{ij}^2)$, where $A$ is the predicted matrix and $UV^\top$ the true low rank matrix. A few example plots can be seen in Figure 3 and a table of results in Table 1. We have implemented the Fast Greedy and Fast Local Search algorithms with 3 inner optimization iterations. In the first few iterations there is a spike in the relative MSE of the algorithms that use the OPTIMIZE_FAST routine. We attribute this to the aggressive alternating minimization steps of this procedure and conjecture that adding a regularization term to the objective might smoothen the spike. However, the Fast Local Search algorithm still gives the best overall performance in terms of how well it approximates the true low rank matrix $UV^\top$, and in particular with a very small rank—practically the same as the true underlying rank.

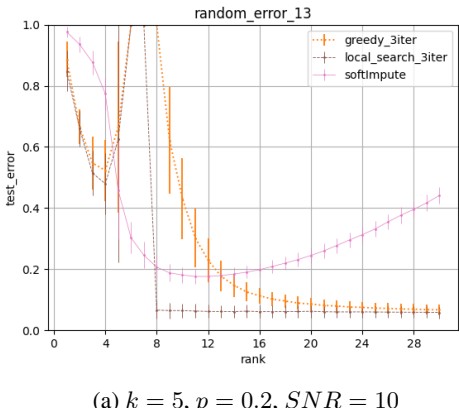
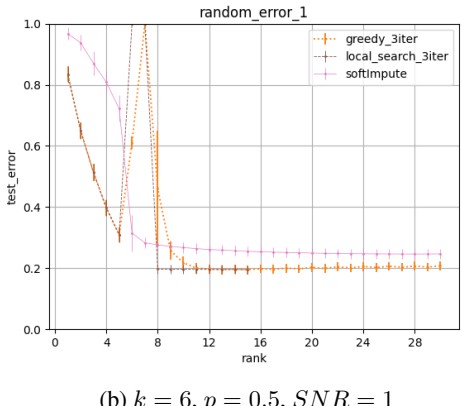

(a) $k = 5$, $p = 0.2$, $SNR = 10$          (b) $k = 6$, $p = 0.5$, $SNR = 1$

Figure 3: Test error vs rank in the matrix completion problem of Section 4.2. Bands of $\pm 1$ standard error are shown. Note that SoftImpute starts to overfit for ranks larger than 12 in (a). The "jumps" at around rank 5-7 happen because of overshooting (taking too large a step) during the insertion the rank-1 component in both Fast Greedy and Fast Local Search. More specifically, these implementations only apply 3 iterations of the inner optimization step, which in some cases are too few to amend the overshooting. However, after a few more iterations of the algorithm the overshooting issue is resolved (i.e. the algorithm has had enough iterations to scale down the rank-1 component that caused the overshooting).

| Algorithm | random_error_13 | random_error_1 | random_error_2 |
|---|---|---|---|
| SoftImpute (Mazumder et al. (2010)) | 0.1759/10 | 0.2465/28 | 0.2284/30 |
| Fast Greedy (Algorithm 2.3) | 0.0673/30 | 0.1948/13 | 0.1826/21 |
| Fast Local Search (Algorithm 5) | 0.0613/14 | 0.1952/15 | 0.1811/15 |

Table 1: Lowest test error for any rank in the matrix completion problem of Section 4.2, and associated rank returned by each algorithm. In the form error/rank.

## 4.3 RECOMMENDER SYSTEMS

In this section we compare our algorithms on the task of movie recommendation on the Movielens datasets Harper & Konstan (2015). In order to evaluate the algorithms, we perform random 80%-20% train-test splits that are the same for all algorithms and measure the mean RMSE in the test set. If we let $\Omega \subseteq [m] \times [n]$ be the set of user-movie pairs in the training set, we assume that the true user-movie matrix is low rank, and thus pose (1) with $R(A) = \frac{1}{2}\|\Pi_\Omega(M - A)\|_F^2$. We make the following slight modification in order to take into account the range of the ratings $[1, 5]$: We clip the entries of $A$ between 1 and 5 when computing $\nabla R(A)$ in Algorithm 2.3 and Algorithm 5. In other words, instead of $\Pi_\Omega(A - M)$ we compute the gradient as $\Pi_\Omega(\text{clip}(A, 1, 5) - M)$. This is similar to replacing our objective by a Huber loss, with the difference that we only do so in the steps that we mentioned and not the inner optimization step, mainly for runtime efficiency reasons.

The results can be seen in Table 2. We do not compare with Fast Local Search, as we found that it only provides an advantage for small ranks ($< 30$), and otherwise matches Fast Greedy. For the inner optimization steps we have used the LSQR algorithm with 2 iterations in the 100K and 1M datasets, and with 3 iterations in the 10M dataset. Note that even though the SVD algorithm by Koren et al. (2009) as implemented by Hug (2020) (with no user/movie bias terms) is a highly tuned algorithm for recommender systems that was one of the top solutions in the famous Netflix prize, it has comparable performance to our general-purpose Algorithm 2.3.

Finally, Table 3 demonstrates the speedup achieved by our algorithms over the basic greedy implementation. It should be noted that the speedup compared to the basic greedy of Shalev-Shwartz et al. (2011) (Algorithm 1) is larger as rank increases, since the fast algorithms scale linearly with rank, but the basic greedy scales quadratically.

| Algorithm | MovieLens 100K | MovieLens 1M | MovieLens 10M |
|---|---|---|---|
| NMF (Lee & Seung (2001)) | 0.9659 | 0.9166 | 0.8960 |
| SoftImpute | 1.0106 | 0.9599 | 0.957 |
| Alternating Minimization | **0.9355** | 0.8732 | 0.8410 |
| SVD (Koren et al. (2009)) | 0.9533 | 0.8743 | **0.8315** |
| Fast Greedy (Algorithm 2.3) | 0.9451 | **0.8714** | 0.8330 |

Table 2: Mean RMSE and standard error among 5 random splits for 100K and 1M with standard errors $< 0.01$, and 3 random splits for 10M with standard errors $< 0.001$. The rank of the prediction is set to 100 except for NMF where it is 15 and Fast Greedy in the 10M dataset where it is chosen to be 35 by cross-validation. Alternating Minimization is a well known algorithm (e.g. Srebro et al. (2004)) that alternatively minimizes the left and right subspace, and also uses Frobenius norm regularization. For SoftImpute and Alternating Minimization we have found the best choice of parameters by performing a grid search over the rank and the multiplier of the regularization term. We have found the best choice of parameters by performing a grid search over the rank and the multiplier of the regularization term. We ran 20 iterations of Alternating Minimization in each case.

| Algorithm | Figure 3 (a) | Movielens 100K | Movielens 1M |
|---|---|---|---|
| SoftImpute | 10.6 | 9.4 | 40.6 |
| Alternating Minimization | 18.9 | 252.0 | 1141.4 |
| Greedy (Shalev-Shwartz et al. (2011)) | 18.8 | 418.4 | 4087.3 |
| Fast Greedy | 10.2 | 43.4 | 244.2 |
| Fast Local Search | 10.8 | 46.1 | 263.0 |

Table 3: Runtimes (in seconds) of different algorithms for fitting a rank=30 solution in various experiments. Code written in python and tested on an Intel Skylake CPU with 16 vCPUs.

It is important to note that our goal here is not to be competitive with the best known algorithms for matrix completion, but rather to propose a general yet practically applicable method for rank-constrained convex optimization. For a recent survey on the best performing algorithms in the Movielens datasets see Rendle et al. (2019). It should be noted that a lot of these algorithms have significant performance boost compared to our methods because they use additional features (meta information about each user, movie, timestamp of a rating, etc.) or stronger models (user/movie biases, "implicit" ratings). A runtime comparison with these recent approches is an interesting avenue for future work. As a rule of thumb, however, Fast Greedy has roughly the same runtime as SVD (Koren et al. (2009)) in each iteration, i.e. $O(|\Omega|r)$, where $\Omega$ is the set of observable elements and $r$ is the rank. As some better performing approaches have been reported to be much slower than SVD (e.g. SVD++ is reported to be 50-100x slower than SVD in the Movielens 100K and 1M datasets (Hug (2020)), this might also suggest a runtime advantage of our approach compared to some better performing methods.

## 5 CONCLUSIONS

We presented simple algorithms with strong theoretical guarantees for optimizing a convex function under a rank constraint. Although the basic versions of these algorithms have appeared before, through a series of careful runtime, optimization, and generalization performance improvements that we introduced, we have managed to reshape the performance of these algorithms in all fronts. Via our experimental validation on a host of practical problems such as low-rank matrix recovery with missing data, robust principal component analysis, and recommender systems, we have shown that the performance in terms of the solution quality matches or exceeds other widely used and even specialized solutions, thus making the argument that our Fast Greedy and Fast Local Search routines can be regarded as strong and practical general purpose tools for rank-constrained convex optimization. Interesting directions for further research include the exploration of different kinds of regularization and tuning for machine learning applications, as well as a competitive implementation and extensive runtime comparison of our algorithms.

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

# A  APPENDIX

## A.1  PRELIMINARIES AND NOTATION

Given an positive integer $k$, we denote $[k] = \{1, 2, \ldots, k\}$. Given a matrix $A$, we denote by $\|A\|_F$ its Frobenius norm, i.e. the $\ell_2$ norm of the entries of $A$ (or equivalently of the singular values of $A$). The following lemma is a simple corollary of the definition of the Frobenius norm:

**Lemma A.1.** *Given two matrices $A \in \mathbb{R}^{m \times n}, B \in \mathbb{R}^{m \times n}$, we have $\|A + B\|_F^2 \leq 2\left(\|A\|_F^2 + \|B\|_F^2\right)$.*

*Proof.*

$$\|A + B\|_F^2 = \sum_{ij} (A + B)_{ij}^2 \leq 2 \sum_{ij} (A_{ij}^2 + B_{ij}^2) = 2(\|A\|_F^2 + \|B\|_F^2)$$

$\square$

**Definition A.2** (Rank-restricted smoothness, strong convexity, condition number). *Given a convex function $R \in \mathbb{R}^{m \times n} \to \mathbb{R}$ and an integer parameter $r$, the* rank-restricted smoothness *of $R$ at rank $r$ is the minimum constant $\rho_r^+ \geq 0$ such that for any two matrices $A \in \mathbb{R}^{m \times n}, B \in \mathbb{R}^{m \times n}$ such that $\mathrm{rank}(A - B) \leq r$, we have*

$$R(B) \leq R(A) + \langle \nabla R(A), B - A \rangle + \frac{\rho_r^+}{2} \|B - A\|_F^2 \,.$$

*Similarly, the* rank-restricted strong convexity *of $R$ at rank $r$ is the maximum constant $\rho_r^- \geq 0$ such that for any two matrices $A \in \mathbb{R}^{m \times n}, B \in \mathbb{R}^{m \times n}$ such that $\mathrm{rank}(A - B) \leq r$, we have*

$$R(B) \geq R(A) + \langle \nabla R(A), B - A \rangle + \frac{\rho_r^-}{2} \|B - A\|_F^2 \,.$$

*Given that $\rho_r^+, \rho_r^-$ exist and are nonzero, the* rank-restricted condition number *of $R$ at rank $r$ is then defined as*

$$\kappa_r = \frac{\rho_r^+}{\rho_r^-}$$

Note that $\rho_r^+$ is increasing and $\rho_r^-$ is decreasing in $r$. Therefore, even though our bounds are proven in terms of the constants $\frac{\rho_1^+}{\rho_r^-}$ and $\frac{\rho_2^+}{\rho_r^-}$, these quantities are always at most $\frac{\rho_r^+}{\rho_r^-} = \kappa_r$ as long as $r \geq 2$, and so they directly imply the same bounds in terms of the constant $\kappa_r$.

**Definition A.3** (Spectral norm). *Given a matrix $A \in \mathbb{R}^{m \times n}$, we denote its spectral norm by $\|A\|_2$. The spectral norm is defined as*

$$\|A\|_2 = \max_{x \in \mathbb{R}^n} \frac{\|Ax\|_2}{\|x\|_2},$$

**Definition A.4** (Singular value thresholding operator). *Given a matrix $A \in \mathbb{R}^{m \times n}$ of rank $k$, a singular value decomposition $A = U\Sigma V^\top$ such that $\Sigma_{11} \geq \Sigma_{22} \geq \cdots \geq \Sigma_{kk}$, and an integer $1 \leq r \leq k$, we define $H_r(A) = U\Sigma'V^\top$, here $\Sigma'$ is a diagonal matrix with*

$$\Sigma'_{ii} = \begin{cases} \Sigma_{ii} & \text{if } i \leq r \\ 0 & \text{otherwise} \end{cases}$$

*In other words, $H_r(\cdot)$ is an operator that eliminates all but the top $r$ highest singular values of a matrix.*

**Lemma A.5** (Weyl's inequality). *For any matrix $A$ and integer $i \geq 1$, let $\sigma_i(A)$ be the $i$-th largest singular value of $A$ or $0$ if $i > \text{rank}(A)$. Then, for any two matrices $A$, $B$ and integers $i \geq 1, j \geq 1$:*

$$\sigma_{i+j-1}(A + B) \leq \sigma_i(A) + \sigma_j(B)$$

A proof of the previous fact can be found e.g. in Fisk (1996).

**Lemma A.6** ($H_r(\cdot)$ optimization problem). *Let $A \in \mathbb{R}^{m \times n}$ be a rank-$k$ matrix and $r \in [k]$ be an integer parameter. Then $M = \frac{1}{\lambda} H_r(A)$ is an optimal solution to the following optimization problem:*

$$\max_{\text{rank}(M) \leq r} \left\{ \langle A, M \rangle - \frac{\lambda}{2} \|M\|_F^2 \right\} \tag{6}$$

*Proof.* Let $U\Sigma V^\top = \sum_i \Sigma_{ii} U_i V_i^\top$ be a singular value decomposition of $A$. We note that (6) is equivalent to

$$\min_{\text{rank}(M) \leq r} \|A - \lambda M\|_F^2 := f(M) \tag{7}$$

Now, note that $f(\frac{1}{\lambda} H_r(A)) = \|A - H_r(A)\|_F^2 = \sum_{i=r+1}^{k} \Sigma_{ii}^2$. On the other hand, by applying Weyl's inequality (Lemma A.5) for $j = r + 1$,

$$f(M) = \|A - \lambda M\|_F^2 = \sum_{i=1}^{k+r} \sigma_i^2(A - \lambda M) \geq \sum_{i=1}^{k+r} (\sigma_{i+r}(A) - \sigma_{r+1}(\lambda M))^2 = \sum_{i=r+1}^{k} \Sigma_{ii}^2,$$

where the last equality follows from the fact that $\text{rank}(A) = k$ and $\text{rank}(M) \leq r$. Therefore, $M = \frac{1}{\lambda} H_r(A)$ minimizes (7) and thus maximizes (6). $\qquad\square$

## A.2    PROOF OF THEOREM 2.1 (GREEDY)

We will start with the following simple lemma about the Frobenius norm of a sum of matrices with orthogonal columns or rows:

**Lemma A.7.** *Let $U \in \mathbb{R}^{m \times r}, V \in \mathbb{R}^{n \times r}, X \in \mathbb{R}^{m \times r}, Y \in \mathbb{R}^{n \times r}$ be such that the columns of $U$ are orthogonal to the columns of $X$ or the columns of $V$ are orthogonal to the columns of $Y$. Then $\|UV^\top + XY^\top\|_F^2 = \|UV^\top\|_F^2 + \|XY^\top\|_F^2$.*

*Proof.* If the columns of $U$ are orthogonal to those of $X$, then $U^\top X = \mathbf{0}$ and if the columns of $V$ are orthogonal to those of $Y$, then $Y^\top V = \mathbf{0}$. Therefore in any case $\langle UV^\top, XY^\top \rangle = \text{Tr}(VU^\top XY^\top) = \text{Tr}(U^\top XY^\top V) = \mathbf{0}$, implying

$$\|UV^\top + XY^\top\|_F^2 = \|UV^\top\|_F^2 + \|XY^\top\|_F^2 + 2\langle UV^\top, XY^\top \rangle = \|UV^\top\|_F^2 + \|XY^\top\|_F^2$$

$\qquad\square$

Additionally, we have the following lemma regarding the optimality conditions of (2):

**Lemma A.8.** *Let $A = UXV^\top$ where $U \in \mathbb{R}^{m \times r}$, $X \in \mathbb{R}^{r \times r}$, and $V \in \mathbb{R}^{n \times r}$, such that $X$ is the optimal solution to (2). Then for any $u \in \text{im}(U)$ and $v \in \text{im}(V)$ we have that $\langle \nabla R(A), uv^\top \rangle = 0$.*

*Proof.* By the optimality condition of 2, we have that

$$U^\top \nabla R(A) V = \mathbf{0}$$

Now, for any $u = Ux$ and $v = Vy$ we have

$$\langle \nabla R(A), uv^\top \rangle = u^\top \nabla R(A) v = x^\top U^\top \nabla R(A) V y = 0$$

$\square$

We are now ready for the proof of Theorem 2.1.

*Proof.* Let $A_{t-1}$ be the current solution $UV^\top$ before iteration $t - 1 \geq 0$. Let $u \in \mathbb{R}^m$ and $v \in \mathbb{R}^m$ be left and right singular vectors of matrix $\nabla R(A)$, i.e. unit vectors maximizing $|\langle \nabla R(A), uv^\top \rangle|$. Let

$$\mathcal{B}_t = \{B | B = A_{t-1} + \eta uv^T, \eta \in \mathbb{R}\}.$$

By smoothness we have

$$
\begin{aligned}
R(A_{t-1}) - R(A_t) &\geq \max_{B \in \mathcal{B}_t} \{R(A_{t-1}) - R(B)\} \\
&\geq \max_{B \in \mathcal{B}_t} \left\{ -\langle \nabla R(A_{t-1}), B - A_{t-1} \rangle - \frac{\rho_1^+}{2} \|B - A_{t-1}\|_F^2 \right\} \\
&\geq \max_\eta \left\{ \eta \langle \nabla R(A_{t-1}), uv^\top \rangle - \eta^2 \frac{\rho_1^+}{2} \right\} \\
&= \max_\eta \left\{ \eta \|\nabla R(A_{t-1})\|_2 - \eta^2 \frac{\rho_1^+}{2} \right\} \\
&= \frac{\|\nabla R(A_{t-1})\|_2^2}{2\rho_1^+}
\end{aligned}
$$

where $\|\cdot\|_2$ is the spectral norm (i.e. maximum magnitude of a singular value).

On the other hand, by strong convexity and noting that

$$\text{rank}(A^* - A_{t-1}) \leq \text{rank}(A^*) + \text{rank}(A_{t-1}) \leq r^* + r,$$

$$R(A^*) - R(A_{t-1}) \geq \langle \nabla R(A_{t-1}), A^* - A_{t-1} \rangle + \frac{\rho_{r+r^*}^-}{2} \|A^* - A_{t-1}\|_F^2. \tag{8}$$

Let $A_{t-1} = UV^\top$ and $A^* = U^* V^{*\top}$. We let $\Pi_{\text{im}(U)} = U(U^\top U)^+ U^\top$ and $\Pi_{\text{im}(V)} = V(V^\top V)^+ V^\top$ denote the orthogonal projections onto the images of $U$ and $V$ respectively. We now write

$$A^* = U^* V^{*\top} = (U^1 + U^2)(V^1 + V^2)^\top = U^1 V^{1\top} + U^1 V^{2\top} + U^2 V^{*\top}$$

where $U^1 = \Pi_{\text{im}(U)} U^*$ is a matrix where every column of $U^*$ is replaced by its projection on $\text{im}(U)$ and $U^2 = U^* - U^1$ and similarly $V^1 = \Pi_{\text{im}(V)} V^*$ is a matrix where every column of $V^*$ is replaced by its projection on $\text{im}(V)$ and $V^2 = V^* - V^1$. By setting $U' = (-U \mid U^1)$ and $V' = (V \mid V^1)$ we can write

$$A^* - A_{t-1} = U' V'^\top + U^1 V^{2\top} + U^2 V^{*\top}$$

where $\text{im}(U') = \text{im}(U)$ and $\text{im}(V') = \text{im}(V)$. Also, note that

$$\text{rank}(U^1 V^{2\top}) \leq \text{rank}(V^2) \leq \text{rank}(V^*) = \text{rank}(A^*) \leq r^*$$

and similarly $\mathrm{rank}(U^2 V^{*\top}) \le r^*$. So now the right hand side of (8) can be reshaped as

$$\langle \nabla R(A_{t-1}), A^* - A_{t-1} \rangle + \frac{\rho^-_{\bar{r}+r^*}}{2} \|A^* - A_{t-1}\|_F^2$$

$$= \langle \nabla R(A_{t-1}), U'V'^\top + U^1 V^{2\top} + U^2 V^{*\top} \rangle + \frac{\rho^-_{\bar{r}+r^*}}{2} \|U'V'^\top + U^1 V^{2\top} + U^2 V^{*\top}\|_F^2$$

Now, note that since by definition the columns of $U'$ are in $\mathrm{im}(U)$ and the columns of $V'$ are in $\mathrm{im}(V)$, Lemma A.8 implies that $\langle \nabla R(A_{t-1}), U'V'^\top \rangle = 0$. Therefore the above is equal to

$$\langle \nabla R(A_{t-1}), U^1 V^{2\top} + U^2 V^{*\top} \rangle + \frac{\rho^-_{\bar{r}+r^*}}{2} \|U'V'^\top + U^1 V^{2\top} + U^2 V^{*\top}\|_F^2$$

$$\ge \langle \nabla R(A_{t-1}), U^1 V^{2\top} \rangle + \langle \nabla R(A_{t-1}), U^2 V^{*\top} \rangle + \frac{\rho^-_{\bar{r}+r^*}}{2} \left( \|U^1 V^{2\top}\|_F^2 + \|U^2 V^{*\top}\|_F^2 \right)$$

$$\ge 2 \min_{\mathrm{rank}(M) \le r^*} \left\{ \langle \nabla R(A_{t-1}), M \rangle + \frac{\rho^-_{\bar{r}+r^*}}{2} \|M\|_F^2 \right\}$$

$$= -2 \frac{\|H_{r^*}(\nabla R(A_{t-1}))\|_F^2}{2\rho^-_{\bar{r}+r^*}}$$

$$\ge -r^* \frac{\|\nabla R(A_{t-1})\|_2^2}{\rho^-_{\bar{r}+r^*}}$$

where the first equality follows by noticing that the columns of $V'$ and $V^1$ are orthogonal to those of $V^2$ and the columns of $U'$ and $U^1$ are orthogonal to those of $U^2$, and applying Lemma A.7. The last equality is a direct application of Lemma A.6 and the last inequality states that the largest squared singular value is not smaller than the average of the top $r^*$ squared singular values. Therefore we have concluded that

$$\|\nabla R(A_{t-1})\|_2^2 \ge \frac{\rho^-_{\bar{r}+r^*}}{r^*} \left( R(A_{t-1}) - R(A^*) \right)$$

Plugging this back into the smoothness inequality, we get

$$R(A_{t-1}) - R(A_t) \ge \frac{1}{2r^* \kappa} (R(A_{t-1}) - R(A^*))$$

or equivalently

$$R(A_t) - R(A^*) \le \left( 1 - \frac{1}{2r^* \kappa} \right) (R(A_{t-1}) - R(A^*)).$$

Therefore after $L = 2r^* \kappa \log \frac{R(A_0) - R(A^*)}{\epsilon}$ iterations we have

$$R(A_T) - R(A^*) \le \left( 1 - \frac{1}{2r^* \kappa} \right)^L (R(A_0) - R(A^*))$$

$$\le e^{-\frac{L}{2r^* \kappa}} (R(A_0) - R(A^*))$$

$$\le \epsilon$$

Since $A_0 = \mathbf{0}$, the result follows. □

### A.3 PROOF OF THEOREM 2.2 (LOCAL SEARCH)

*Proof.* Similarly to Section A.3, we let $A_{t-1}$ be the current solution before iteration $t - 1 \ge 0$. Let $u \in \mathbb{R}^m$ and $v \in \mathbb{R}^m$ be left and right singular vectors of matrix $\nabla R(A)$, i.e. unit vectors maximizing $|\langle \nabla R(A), uv^\top \rangle|$ and let

$$\mathcal{B}_t = \{B | B = A_{t-1} + \eta uv^T - \sigma_{\min} xy^\top, \eta \in \mathbb{R}\},$$

where $\sigma_{\min}xy^\top = A_{t-1} - H_{r-1}(A_{t-1})$ is the rank-1 term corresponding to the minimum singular value of $A_{t-1}$. By smoothness we have

$$
\begin{aligned}
&R(A_{t-1}) - R(A_t) \\
&\geq \max_{B \in \mathcal{B}_t} \{R(A_{t-1}) - R(B)\} \\
&\geq \max_{B \in \mathcal{B}_t} \left\{ -\langle \nabla R(A_{t-1}), B - A_{t-1} \rangle - \frac{\rho_2^+}{2} \|B - A_{t-1}\|_F^2 \right\} \\
&= \max_{\eta \in \mathbb{R}} \left\{ -\langle \nabla R(A_{t-1}), \eta u v^\top - \sigma_{\min}xy^\top \rangle - \frac{\rho_2^+}{2} \|\eta u v^\top - \sigma_{\min}xy^\top\|_F^2 \right\} \\
&\geq \max_{\eta \in \mathbb{R}} \left\{ -\langle \nabla R(A_{t-1}), \eta u v^\top \rangle - \eta^2 \rho_2^+ - \sigma_{\min}^2 \rho_2^+ \right\} \\
&= \max_{\eta \in \mathbb{R}} \left\{ \eta \|\nabla R(A_{t-1})\|_2 - \eta^2 \rho_2^+ - \sigma_{\min}^2 \rho_2^+ \right\} \\
&= \frac{\|\nabla R(A_{t-1})\|_2^2}{4\rho_2^+} - \sigma_{\min}^2 \rho_2^+ ,
\end{aligned}
$$

where in the last inequality we used the fact that $\langle \nabla R(A_{t-1}), xy^\top \rangle = 0$ following from Lemma A.8, as well as Lemma A.1.

On the other hand, by strong convexity,

$$
R(A^*) - R(A_{t-1}) \geq \langle \nabla R(A_{t-1}), A^* - A_{t-1} \rangle + \frac{\rho_{r+r^*}^-}{2} \|A^* - A_{t-1}\|_F^2 .
$$

Let $A_{t-1} = UV^\top$ and $A^* = U^*V^{*\top}$. We write

$$
A^* = U^*V^{*\top} = (U^1 + U^2)(V^1 + V^2)^\top = U^1V^{1\top} + U^1V^{2\top} + U^2V^{*\top}
$$

where $U^1$ is a matrix where every column of $U^*$ is replaced by its projection on $\operatorname{im}(U)$ and $U^2 = U^* - U^1$ and similarly $V^1$ is a matrix where every column of $V^*$ is replaced by its projection on $\operatorname{im}(V)$ and $V^2 = V^* - V^1$. By setting $U' = (-U \mid U^1)$ and $V' = (V \mid V^1)$ we can write

$$
A^* - A_{t-1} = U'V'^\top + U^1V^{2\top} + U^2V^{*\top}
$$

where $\operatorname{im}(U') = \operatorname{im}(U)$ and $\operatorname{im}(V') = \operatorname{im}(V)$. Also, note that

$$
\operatorname{rank}(U^1V^{2\top}) \leq \operatorname{rank}(V^2) \leq \operatorname{rank}(V^*) = \operatorname{rank}(A^*) \leq r^*
$$

and similarly $\operatorname{rank}(U^2V^{*\top}) \leq r^*$. So we now have

$$
\begin{aligned}
&\langle \nabla R(A_{t-1}), A^* - A_{t-1} \rangle + \frac{\rho_{r+r^*}^-}{2} \|A^* - A_{t-1}\|_F^2 \\
&= \langle \nabla R(A_{t-1}), U'V'^\top + U^1V^{2\top} + U^2V^{*\top} \rangle + \frac{\rho_{r+r^*}^-}{2} \|U'V'^\top + U^1V^{2\top} + U^2V^{*\top}\|_F^2 \\
&= \langle \nabla R(A_{t-1}), U^1V^{2\top} + U^2V^{*\top} \rangle + \frac{\rho_{r+r^*}^-}{2} \|U'V'^\top + U^1V^{2\top} + U^2V^{*\top}\|_F^2 \\
&= \langle \nabla R(A_{t-1}), U^1V^{2\top} + U^2V^{*\top} \rangle + \frac{\rho_{r+r^*}^-}{2} \left( \|U'V'^\top\|_F^2 + \|U^1V^{2\top}\|_F^2 + \|U^2V^{*\top}\|_F^2 \right) \\
&\geq \langle \nabla R(A_{t-1}), U^1V^{2\top} \rangle + \langle \nabla R(A_{t-1}), U^2V^{*\top} \rangle + \frac{\rho_{r+r^*}^-}{2} \left( \|U^1V^{2\top}\|_F^2 + \|U^2V^{*\top}\|_F^2 \right) \\
&\quad + \frac{\rho_{r+r^*}^-}{2} \|U'V'^\top\|_F^2 \\
&\geq 2 \min_{\operatorname{rank}(M) \leq r^*} \left\{ \langle \nabla R(A_{t-1}), M \rangle + \frac{\rho_{r+r^*}^-}{2} \|M\|_F^2 \right\} + \frac{\rho_{r+r^*}^-}{2} \|U'V'^\top\|_F^2 \\
&= -2 \frac{\|H_{r^*}(\nabla R(A_{t-1}))\|_F^2}{2\rho_{r+r^*}^-} + \frac{\rho_{r+r^*}^-}{2} \|U'V'^\top\|_F^2 \\
&\geq -r^* \frac{\|\nabla R(A_{t-1})\|_2^2}{\rho_{r+r^*}^-} + \frac{\rho_{r+r^*}^-}{2} \|U'V'^\top\|_F^2
\end{aligned}
$$

where the second equality follows from the fact that $\langle \nabla R(A_{t-1}), uv^\top \rangle = 0$ for any $u \in \mathrm{im}(U), v \in \mathrm{im}(V)$, the third equality from the fact that $\mathrm{im}(U^2) \perp \mathrm{im}(U') \cup \mathrm{im}(U^1)$ and $\mathrm{im}(V^2) \perp \mathrm{im}(V')$ and by applying Lemma A.7, and the last inequality from the fact that the largest squared singular value is not smaller than the average of the top $r^*$ squared singular values. Now, note that since $\mathrm{rank}(U^1 V^{1\top}) \leq r^* < r = \mathrm{rank}(UV^\top)$,

$$
\begin{aligned}
\|U'V'^\top\|_F^2 &= \|U^1 V^{1\top} - UV^\top\|_F^2 \\
&= \sum_{i=1}^{r} \sigma_i^2(U^1 V^{1\top} - UV^\top) \\
&\geq \sum_{i=1}^{r} (\sigma_{i+r^*}(UV^\top) - \sigma_{r^*+1}(U^1 V^{1\top}))^2 \\
&= \sum_{i=r^*+1}^{r} \sigma_i^2(UV^\top) \\
&\geq (r - r^*)\sigma_{\min}^2(UV^\top) \\
&= (r - r^*)\sigma_{\min}^2(A_{t-1}),
\end{aligned}
$$

where we used the fact that $\mathrm{rank}(U^1 V^{1\top}) \leq r^*$ together with Lemma A.5. Therefore we have concluded that

$$
\|\nabla R(A_{t-1})\|_2^2 \geq \frac{\rho_{r+r^*}^-}{r^*}(R(A_{t-1}) - R(A^*)) + \frac{(\rho_{r+r^*}^-)^2(r - r^*)}{2r^*}\sigma_{\min}^2
$$

Plugging this back into the smoothness inequality and setting $\widetilde{\kappa} = \frac{\rho_2^+}{\rho_{r+r^*}^-}$, we get

$$
\begin{aligned}
R(A_{t-1}) - R(A_t) &\geq \frac{1}{4r^*\widetilde{\kappa}}(R(A_{t-1}) - R(A^*)) + \left(\frac{\rho_{r+r^*}^-(r - r^*)}{8r^*\widetilde{\kappa}} - \rho_2^+\right)\sigma_{\min}^2(A_{t-1}) \\
&\geq \frac{1}{4r^*\widetilde{\kappa}}(R(A_{t-1}) - R(A^*))
\end{aligned}
$$

as long as $r \geq r^*(1 + 8\widetilde{\kappa}^2)$, or equivalently,

$$
R(A_t) - R(A^*) \leq \left(1 - \frac{1}{4r^*\widetilde{\kappa}}\right)(R(A_{t-1}) - R(A^*)).
$$

Therefore after $L = 4r^*\widetilde{\kappa} \log \frac{R(A_0) - R(A^*)}{\epsilon}$ iterations we have

$$
\begin{aligned}
R(A_T) - R(A^*) &\leq \left(1 - \frac{1}{4r^*\widetilde{\kappa}}\right)^L (R(A_0) - R(A^*)) \\
&\leq e^{-\frac{L}{4r^*\widetilde{\kappa}}}(R(A_0) - R(A^*)) \\
&\leq \epsilon
\end{aligned}
$$

Since $A_0 = \mathbf{0}$ and $\widetilde{\kappa} \leq \kappa_{r+r^*}$, the result follows. □

### A.4 TIGHTNESS OF THE ANALYSIS

It is important to note that the $\kappa_{r+r^*}$ factor that appears in the rank bounds of both Theorems 2.1 and 2.2 is inherent in these algorithms and not an artifact of our analysis. In particular, such lower bounds based on the restricted condition number have been previously shown for the problem of sparse linear regression. More specifically, Foster et al. (2015) showed that there is a family of instances in which the analogues of Greedy and Local Search for sparse optimization require the sparsity to be $\Omega(s^*\kappa')$ for constant error $\epsilon > 0$, where $s^*$ is the optimal sparsity and $\kappa'$ is the *sparsity*-restricted condition number. These instances can be easily adjusted to give a *rank* lower bound of $\Omega(r^*\kappa_{r+r^*})$ for constant error $\epsilon > 0$, implying that the $\kappa$ dependence in Theorem 2.1 is tight for Greedy. Furthermore, specifically for Local Search, Axiotis & Sviridenko (2020) additionally

showed that there is a family of instances in which the analogue of Local Search for sparse optimization requires a sparsity of $\Omega(s^*(\kappa')^2)$. Adapting these instances to the setting of rank-constrained convex optimization is less trivial, but we conjecture that it is possible, which would lead to a rank lower bound of $\Omega(r^*\kappa_{r+r^*}^2)$ for Local Search.

We present the following lemma, which essentially states that sparse optimization lower bounds for Orthogonal Matching Pursuit (OMP, Pati et al. (1993)) (resp. Orthogonal Matching Pursuit with Replacement (OMPR, Jain et al. (2011))) in which the optimal sparse solution is also a global optimum, immediately carry over (up to constants) to rank-constrained convex optimization lower bounds for Greedy (resp. Local Search).

**Lemma A.9.** *Let $f \in \mathbb{R}^n \to \mathbb{R}$ and $x^* \in \mathbb{R}^n$ be an $s^*$-sparse vector that is also a global minimizer of $f$. Also, let $f$ have restricted smoothness parameter $\beta$ at sparsity level $s + s^*$ for some $s \geq s^*$ and restricted strong convexity parameter $\alpha$ at sparsity level $s + s^*$. Then we can define the rank-constrained problem, with $R : \mathbb{R}^{n \times n} \to \mathbb{R}$,*

$$\min_{\text{rank}(A) \leq s^*} R(A) := f(\text{diag}(A)) + \frac{\beta}{2}\|A - \text{diag}(A)\|_F^2, \tag{9}$$

*where $\text{diag}(A)$ is a vector containing the diagonal of $A$. $R$ has rank-restricted smoothness at rank $s + s^*$ at most $2\beta$ and rank-restricted strong convexity at rank $s + s^*$ at least $\alpha$. Suppose that we run $t$ iterations of OMP (resp. OMPR) starting from a solution $x$, to get solution $x'$, and similarly run $t$ iterations of Greedy (resp. Local Search) starting from solution $A = \text{diag}(x)$ (where $\text{diag}(x)$ is a diagonal matrix with $x$ on the diagonal) to get solution $A'$. Then $A'$ is diagonal and $\text{diag}(A') = x'$. In other words, in this scenario OMP and Greedy (resp. OMPR and Local Search) are equivalent.*

*Proof.* Note that for any solution $\widehat{A}$ of $R$ we have $R(\widehat{A}) \geq f(\text{diag}(\widehat{A})) \geq f(x^*)$, with equality only if $\widehat{A}$ is diagonal. Furthermore, $\text{rank}(\text{diag}(x^*)) \leq s^*$, meaning that $\text{diag}(x^*)$ is an optimal solution of (9). Now, given any diagonal solution $A$ of (9) such that $A = \text{diag}(x)$, we claim that one step of either Greedy or Local Search keeps it diagonal. This is because

$$\nabla R(A) = \text{diag}(\nabla f(x)) + \frac{\beta}{2}(A - \text{diag}(A)) = \text{diag}(\nabla f(x)).$$

Therefore the largest eigenvalue of $\nabla R(A)$ has corresponding eigenvector $\mathbf{1}_i$ for some $i$, which implies that the rank-1 component which will be added is a multiple of $\mathbf{1}_i \mathbf{1}_i^\top$. For the same reason the rank-1 component removed by Local Search will be a multiple of $\mathbf{1}_j \mathbf{1}_j^\top$ for some $j$. Therefore running Greedy (resp. Local Search) on such an instance is identical to running OMP (resp. OMPR) on the diagonal. $\square$

Together with the lower bound instances of Foster et al. (2015) (in which the global minimum property is true), it immediately implies a rank lower bound of $\Omega(r^*\kappa_{r+r^*})$ for getting a solution with constant error for rank-constrained convex optimization. On the other hand, the lower bound instances of Axiotis & Sviridenko (2020) give a *quadratic* lower bound in $\kappa$ for OMPR. The above lemma cannot be directly applied since the sparse solutions are not global minima, but we conjecture that a similar proof will give a rank lower bound of $\Omega(r^*\kappa_{r+r^*}^2)$ for rank-constrained convex optimization with Local Search.

## A.5   ADDENDUM TO SECTION 4

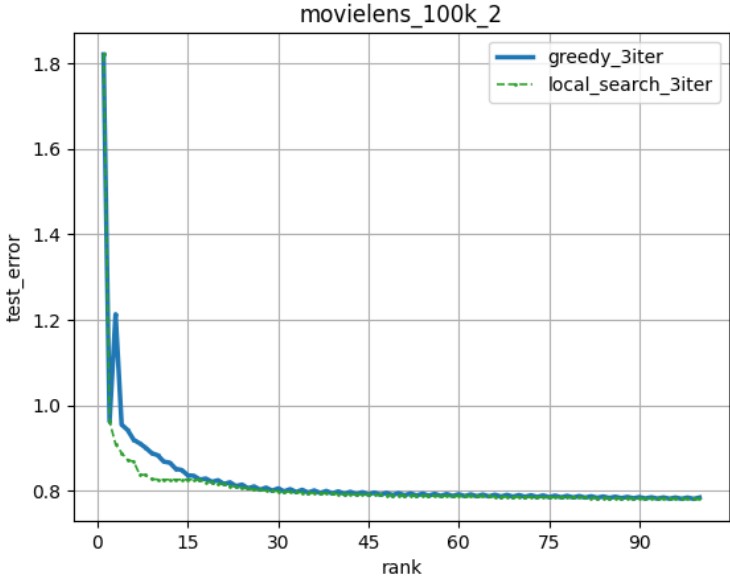

Figure 4: One of the splits of the Movielens 100K dataset. We can see that for small ranks the Fast Local Search solution is better and more stable, but for larger ranks it does not provide any improvement over the Fast Greedy algorithm.

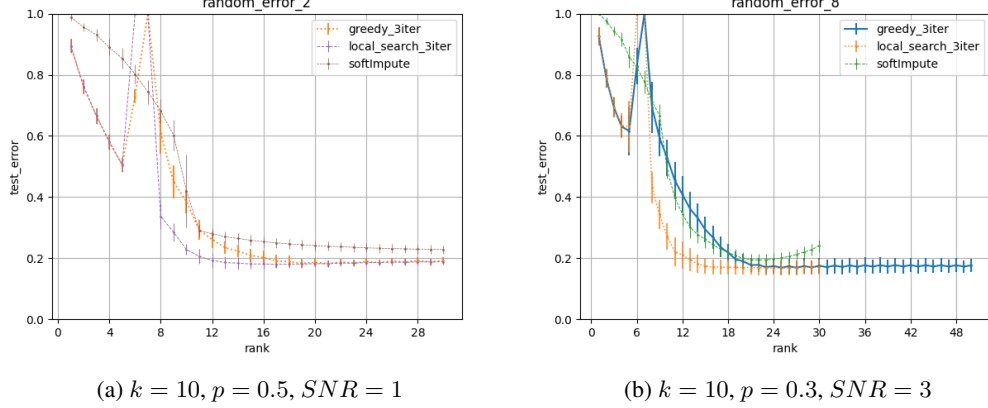

(a) $k = 10, p = 0.5, SNR = 1$             (b) $k = 10, p = 0.3, SNR = 3$

Figure 5: Test error vs rank in the matrix completion problem of Section 4.2. Bands of $\pm 1$ standard error are shown.

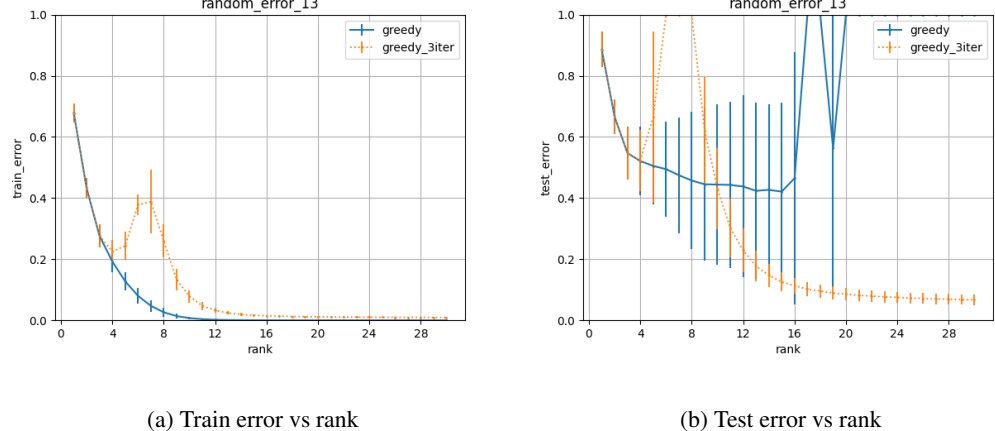

(a) Train error vs rank

(b) Test error vs rank

Figure 6: Performance of greedy with fully solving the inner optimization problem (left) and applying 3 iterations of the LSQR algorithm (right) in the matrix completion problem of Section 4.2. $k = 5$, $p = 0.2$, $SNR = 10$. Bands of $\pm 1$ standard error are shown. This experiment shows why it is crucial to apply some kind of regularization to the Fast Greedy and Fast Local Search algorithms for machine learning applications.

