# OpenReview forum: "Local Search Algorithms for Rank-Constrained Convex Optimization"
_ICLR.cc/2021/Conference — ICLR 2021 Poster_

### Official Review · AnonReviewer4 · 2020-10-20
**A rather terse paper with a few high-level or intuitive justification**

**Rating:** 6
**Confidence:** 3

**Review:**

This work deals with minimizing a convex loss under a rank constraint. Relatively tight bounds and numerical tests are presented.

i) Overall, this is a very terse paper at least to appreciate its analytical merits. A couple of theorems are presented in the main text without elaboration, which makes it hard to gain insights. For example, it is claimed that the analysis of Theorem 1 improves over [Shalev-Shwartz 2011], but it is not clear where the improvement comes from. Is it a new inequality or something else? In addition, in the local search of Algorithm 2, it is also difficult to grasp intuitively why it offers an improvement over Algorithm 1.

ii) In addition, some parameters which seem important in the analysis are not explained well. For example, in both Theorems 1 and 2 the phrase “let A* be the optimal solution,” implies that the solution to (1) is unique. But this has not been established. And for the rank-restricted condition number \kappa_r, one wonders what will happen if \rho_r^- = 0. In this case, it seems that \kappa_r = \infty, and the results in Theorems 1 and 2 no longer hold.

iii) It seems that in order for Theorem 1 to hold, r^* (which is the constraint on rank(A)), has to be smaller than 0.5min{m,n}? What if this condition is not met?

iv) The numerical tests demonstrate the efficiency of the proposed algorithms (and its fast implementation), but they are not supportive enough for the theoretical findings.

v) Why does the test error of soft impute (the pink line) increase with rank in Fig.3 (a)? And no clear-enough description of implementation is provided so one has to check the code carefully.

vi) In Section 4.1, it appears that the method running fewer iterations for the subproblem can be also theoretically supported by solving this subproblem to a certain accuracy.

The current form of this paper do not cross the acceptance threshold of this competitive conference. A major revision is due to better argue and justify the claims, but also provide enough implementation details.

---

> ### Author Response · Authors · 2020-11-24
> **Author response**
>
> We thank the reviewer for the very detailed comments. We address each one of them below:
>
> **i)**  We have added the following high-level explanation in the manuscript in order to explain the theoretical improvements compared to previous work, as well as the advantage of local search over greedy:
>
> "The main bottleneck in Shalev-Shwartz et al. (2011) is the fact that the analysis is done in terms of the squared nuclear norm of the optimal solution. As the worst-case discrepancy between the squared nuclear norm and the rank is $R(0)/\epsilon$, their bounds inherit this factor. Our analysis works directly with the rank, in the spirit of sparse optimization results (e.g. Shalev-Shwartz et al. (2010), Jain et al. (2014), and Axiotis & Sviridenko (2020)). A challenge compared to these works is the need for a suitable notion of "intersection" between two sets of vectors. The main technical contribution of this work is to show that the orthogonal projection of one set of vectors into the span of the other is such a notion, and, based on this, to define a decomposition of the optimal solution that is used in the analysis."
>
> "One drawback of Algorithm 1 is that it increases the rank in each iteration. Algorithm 2 is a modification of Algorithm 1, in which the rank is truncated in each iteration. The advantage of Algorithm 2 compared to Algorithm 1 is that it is able to make progress without increasing the rank of A, while Algorithm 1 necessarily increases the rank in each iteration. More specifically, because of the greedy nature of Algorithm 1, some rank-1 components that have been added to A might become obsolete or have reduced benefit after a number of iterations. Algorithm 2 is able to identify such candidates and remove them, thus allowing it to continue making progress."
>
>
> **ii)** We agree with the reviewer that the optimal solution is not necessarily unique. This was a typo on our behalf, and we have changed it to "let $A^*$ be any fixed optimal solution" in the statements of Theorem 1 and Theorem 2. Regarding what will happen if $\rho^- = 0$ (i.e. the restricted condition number is infinite), this would imply that $f$ does not have the restricted strong convexity property, however this property is assumed to be true for all our theorems to hold.
>
>
> **iii)** As any matrix $A$ has rank at most $\min (m,n)$, in this case any solution to $\min_A R(A)$ gives a $2$-approximation of the target rank (i.e. $r \leq 2r^*$). Therefore this is a relatively easy case, and the harder regime is when $r^* \ll \min(m,n)$.
>
>
> **iv)** We thank the reviewer for the honest opinion, although we are not sure if we understand the criticism here. To elaborate on our approach, the goal of our paper is to first present and theoretically ground our algorithms in a standard setting (in fact improving over previous theoretical bounds for the same setting), and then to propose practical versions of these algorithms, together with experiments (including real data) that justify their competitive performance in practice. As has also been observed in sparse linear regression, good theoretical performance based on the restricted condition number usually is a good indicator for good performance in practical instances. For instance it is trivial to set the restricted condition number to infinity in sparse linear regression, by duplicating one of the feature vectors, but the restricted condition number is still used to accurately benchmark theoretical results.
>
>
> **v)** The test error of softImpute increases because it overfits to the training data. As the rank of $A$ increases, it is able to fit to the training data better. There is a rank threshold after which the test error starts increasing and the train error keeps decreasing.
>
>
> **vi)** We thank the reviewer for the thoughtful suggestion. We agree that it is probably possible to theoretically justify solving the inner problem to a certain accuracy as opposed to exactly. We assumed that the inner solver is exact, as this makes the statement of the theorems and the presentation of the proof cleaner.

---

### Official Review · AnonReviewer1 · 2020-10-29
**Review for Paper2635**

**Rating:** 7
**Confidence:** 3

**Review:**

This paper studies the problem of rank-constrained convex optimization. Its primary contributions are an improved analysis with convergence rate logarithmic in R(0)/eps (rather than poly(R(0)/eps)), as well as some adjustments to the implementations of the greedy and local search methods which result in better empirical performance.

In terms of experimental performance, the authors consider several problem settings, including robust PCA, matrix completion with random noise, and recommender systems. Here, they show that their methods are generally competitive with, and sometimes even improve upon previous methods both in performance and in runtime.

One clarification: is the Greedy algorithm (Algorithm 1, with Optimize rather than Optimize_Fast) exactly the same as in Shalev-Shwartz et al. (2011)? If so, this could be made more clear, especially since the implication is that the benefit for this case appears to be just in terms of the improved analysis for a logarithmic dependence. Building on this, it would be nice to say just a bit more about how (at a high level) the analysis has changed to allow the exponential improvement, as well as how these changes compare to the analogous results of, e.g., Axiotis & Sviridenko (2020).

Overall, I feel the paper provides a nice contribution, both theoretical and practical, to the problem of rank-constrained convex optimization that neatly builds upon previous work.


[Minor Comments]

_ "we will found useful" -> "we will find useful"

_ The abstract claims that the theoretical analysis is "tight". By this do you mean tight in terms of what can be shown (or can hope to be shown) for greedy and/or local search methods, or do you mean tight in an oracle sense? Can either of these be shown?

---

> ### Author Response · Authors · 2020-11-24
> **Author response**
>
> We thank the reviewer for the thoughtful review and suggestions. We address the comments below.
>
> **(i)** Regarding the similarity of Algorithm 1 with Shalev-Shwartz et al. (2011):
>
> The reviewer is correct that the Greedy algorithm (Algorithm 1) is identical to the algorithm that is analyzed in Shalev-Shwartz et al. (2011). Our technical contribution is not on designing this algorithm but on proving that the $\frac{R(0)}{\epsilon}$ factor in the analysis can be replaced by $\log \frac{R(0)}{\epsilon}$. We have made this clearer in the updated version of the manuscript that we will upload soon.
>
>
> **(ii)** Providing intuition on the technical improvement:
>
> We have added the following paragraph in order to give intuition about the technical improvement:
>
> "The main bottleneck in Shalev-Shwartz et al. (2011) is the fact that the analysis is done in terms of the squared nuclear norm of the optimal solution. As the worst-case discrepancy between the squared nuclear norm and the rank is $R(0)/\epsilon$, their bounds inherit this factor. Our analysis works directly with the rank, in the spirit of sparse optimization results (e.g. Shalev-Shwartz et al. (2010), Jain et al. (2014), and Axiotis & Sviridenko (2020)). A challenge compared to these works is the need for a suitable notion of "intersection" between two sets of vectors. The main technical contribution of this work is to show that the orthogonal projection of one set of vectors into the span of the other is such a notion, and, based on this, to define a decomposition of the optimal solution that is used in the analysis."
>
> Regarding the technique of Axiotis & Sviridenko (2020) to obtain linear dependence on the condition number, applying it on this setting would be significantly non-trivial and it is unclear (albeit very interesting) whether it is possible to do so.
>
> **(iii)** Tightness of the analysis:
>
> We thank the reviewer for the very important comment. Here we are referring to the tightness of the analysis. In particular, Theorem 2.1 states that Greedy returns a solution of rank $O(r^* \kappa \log \frac{R(0)}{\epsilon})$ and arbitrarily small error $\epsilon$, and Theorem 2.2 states that given any initial solution of rank $\Theta(r^* \kappa^2)$, Local Search recovers a solution with arbitrarily small error $\epsilon$. What we can show is that there exist instances in which the factor of $\kappa$ is necessary in the analysis of both algorithms.
>
> This means that there are explicit instances in which setting the rank to $O(r^* \kappa^{1-\delta} \log \frac{R(0)}{\epsilon})$ for any $\delta > 0$ leads to inability for either Greedy or Local Search to converge to an arbitrarily small error.
>
> As a matter of fact, for the case of Local Search applied to sparse linear regression (generally a much simpler problem than rank-constrained convex optimization) there are explicit instances showing that even the $\kappa^2$ factor is tight. While crafting a response to this review, we have realized that, maybe counter-intuitively, these instances don't trivially generalize to the case of rank-constrained optimization. We conjecture that this lower bound of $\kappa^2$ does transfer to this setting, but it might take some work to craft the specific instances. We have removed the word "tight" from the abstract and instead added a section to explain these instances and clarify these points

---

### Official Review · AnonReviewer3 · 2020-11-01
**review of "Local Search Algorithms for Rank-Constrained Convex Optimization"**

**Rating:** 7
**Confidence:** 5

**Review:**

Paper is about the problem of minimizing a convex function R(A) subject to a rank constraint rank(A)<=r

Paper builds on an algorithm (GECO, for Greedy Efficient Component Optimization) proposed by Shalev-Shwartz et al. (2011), which was already proven to find a solution to the problem in:

O(r* kappa_{r+r*} R(0) / epsilon) where kappa is related to the function's condition number.

The complexity of the solution of GECO is improved by the author of the present paper, by using techniques borrowed from sparse convex optimization, and they get a dependence in R(0) / epsilon that is logarithmic. This is their first contribution, in Theorem 2.1.

Second contribution is to use a series of approximations to the different steps involved in GECO. Using the proposed approximations, authors prove, in Theorem 2.2, that the complexity can further be reduced to


O(r* kappa_{r+r*} log { [ R(0) - R(A*) ] / epsilon})

These analyses and the experimental results are encouraging and make the paper be a good contribution.

I have a couple of questions regarding numerical experiments:

1 - I could not figure out why in Figure 3, the orange plots (greedy_3iter) have a jump in test error around Rank ~ 7

2 - Table 3 gives runtimes of the proposed algorithms versus the parent algorithm GECO. The proposed algorithm runs faster, OK, but does it have the same or lower reconstruction / test error? In fact, what is missing is test error of GECO in table 2. I fear that it is not reported because it was better than the proposed algorithm??

Minor: thanks for sharing the source for the paper. The python code really looks like MATLAB though :-(

---

> ### Author Response · Authors · 2020-11-24
> **Author response**
>
> We thank the reviewer for the very thoughtful comments.
>
> **1** - This jump happens because of overshooting (taking too large a step) during the insertion the rank-1 component $uv^\top$. It does not happen in the standard implementation (Algorithm 1), because right after the insertion step there is a full optimization step (line 9 of Algorithm 1) which finds the best combination of current vectors (and, as a result, this step might scale down $uv^\top$ to avoid overshooting). On the other hand, the practical implementation (Fast Greedy) only applies 3 iterations of the inner optimization step, which in some cases are too few to amend the overshooting. However, after a few more iterations of the algorithm the issue is resolved (in some sense, the algorithm has had enough "time" to scale down the rank-1 component that caused the overshooting). We have incorporated this explanation into the text in order to clarify this important observation.
>
> **2** - Based on the plots in Shalev-Shwartz et al. (2010), the test errors for Movielens 100K, 1M, and 10M are (very roughly) 0.984, 0.882, and 0.842 respectively, while the performance of the Fast Greedy is 0.9451, 0.8714, 0.8330 (lower is better). Even though all three of these errors are higher than the test errors of Fast Greedy, we chose not to include these results in Table 2 since we were not able to determine the exact train-test split that was used, or find the source code in order to generate the results. In our own implementation of GECO we were not able to achieve the aforementioned test errors probably due to important implementation differences.  We can include these test errors in our final version of the paper if the reviewer believes that will be helpful.

---

### Official Review · AnonReviewer2 · 2020-11-02
**A greedy approach to solving rank-constrained convex optimization with theoretical guarantees**

**Rating:** 6
**Confidence:** 3

**Review:**

Summary of review:

This paper considers solving rank-constrained convex optimization. This is a fairly general problem that contains several special cases such as matrix completion and robust PCA. This paper presents a local search approach along with an interesting theoretical analysis of their approach. Furthermore, this paper provided extensive simulations to validate their approach. Overall, the paper provided solid justification for their approach.

Approach:

The proposed approaches, namely greedy and local search, iteratively add a rank-1 update to the current solution, where the rank-1 update comes from the maximum singular value and eigenvector of the current gradient. After performing the rank-1 update, an inner optimization problem is performed and operates in a low-rank (compared to the dimension of the input) space.

- In the greedy approach, the rank-1 update is simply added to the iterate.
- In the local search approach, the iterate goes through an additional truncation step, which reduces its rank by one before adding the rank-1 update.

Theoretical analysis:

For both approaches, this paper proves that the iterative procedure converges with a solution within $\epsilon$ to the optimum in $r^{\star} \kappa^2 \log{O(1 / \epsilon}$ iterations, where $r^{\star}$ is the rank of the optimal solution of the minimization problem and $\kappa$ is a certain rank-restricted condition number. This result is quite interesting because it provides an explicit upper bound on the rank of the converged iterates. The arguments in the analysis look sound to me.

Validation:

The authors went on to validate their proposed approaches in matrix completion and robust PCA.

For matrix completion, the authors compared their approach to SoftImpute (Mazumder et al 2010), which is a well-known approach in this literature. They showed their approach outperforms SoftImpute in simulations. Then, the authors compared their approach to NMF and SVD on the MovieLens datasets and showed their approach achieves comparable test loss while speeding up the computation by several factors.

Writing:

Overall, the writing is clear and easy to follow. I have several detailed comments below.

Questions:

It would help improve the reviewer's understanding if the author(s) address the following questions in their rebuttable.

--- Note: Properly addressing the questions below is required for the reviewer to better appreciate your results.

(i) In Table 2 and 3, could you add a comparison to SoftImpute and Alternating Minimization with L2 Regularization (say you choose the rank via cross-validation)? How would your results compare to these approaches?

(ii) Specific to the MovieLens benchmark, could you discuss how your approach compares to more recent approaches (as opposed to 2009, https://paperswithcode.com/sota/collaborative-filtering-on-movielens-10m)? Particularly the runtime improvements since these look like the key claims of this experiment. While these approaches may not apply to generic constrained convex optimization, it is still worth discussing?

(iii) How large should I expect the rank-restricted condition number to be in Theorem 2.1 and 2.2? For example, how large are they in the setting of Figure 1 and 3?

Detailed comments:

- Section 2.1 and 2.2: these two theorems and the algorithms came out all of a sudden with no description. Could you add some explanation and describe some intuition?

- Typo, P2: "we will found useful" -> we will find useful

---

> ### Comment · AnonReviewer2 · 2020-11-24
> **Author response?**
>
> Dear Authors,
>
> Could you respond to my questions written in the original review? I believe that the discussion period will finish soon. Your response could help the reviewer better understand your contribution.

---

> > ### Author Response · Authors · 2020-11-24
> > **Response**
> >
> > We thank the reviewer for their patience and are sorry for any inconvenience. We are planning to release our responses very soon.

---

> ### Author Response · Authors · 2020-11-24
> **Author response (1/2)**
>
> We thank the reviewer for the helpful and to-the-point feedback. We will soon be uploading an updated version of the manuscript to reflect the changes that resulted from the comments in this review.
>
> **(i)** We thank the reviewer for pointing this out, we have now compared our approach with SoftImpute and Alternating Minimization with L2 Regularization, as suggested.
>
> For Alternating Minimization with L2 Regularization, we found that in terms of test error it outperforms Fast Greedy on the Movielens 100K dataset, and Fast Greedy outperforms it on the Movielens 1M and 10M datasets. In terms of runtime, Fast Greedy is significantly (5x) faster at recovering a solution of the same rank. One reason for this is that the regularization term $\|M\|_F^2$ is applied on the full matrix $M$, and as a result a full-sized linear system has to be solved in each iteration. On the other hand, the size of our linear system is proportional to the number of observable entries (and not the size of the matrix), and we additionally only run a few iterations of this linear system.
>
> Comparing with SoftImpute, we found that it is significantly faster than Fast Greedy at recovering a solution of the same rank (5-6x), but the improved runtime is at the expense of much higher test error (e.g. RMSE=1 vs 0.945 in the Movielens 100K and 0.960 vs 0.871 in the Movielens 1M dataset).
>
> We have updated Tables 2 and 3 with these new experimental results.
>
>
> **(ii)** We agree with the reviewer that more recent approaches for matrix completion on the Movielens dataset should be mentioned. It is important to note that a qualitative difference between these best performing approaches and ours is that they either use additional features (meta information about each user, movie, timestamp of a rating, etc.) or a stronger model (user/movie biases, "implicit" ratings). Without a doubt, these algorithms give a much better test error than our approach. Unfortunately, we were not able to meaningfully compare runtimes experimentally due to the difference in programming languages with the best performing methods (C vs Python) which introduces inherent performance differences. However, we expect the best performing methods to be significantly slower than ours. In particular, we expect Fast Greedy to be roughly of the same runtime magnitude as SVD (Koren et al. 2009), since each iteration in practice runs in time $O(|\Omega| r)$, where $\Omega$ is the set of observable entries and $r$ is the rank. But SVD++, which is one of the best performing methods, is reported to run 50-100 times slower than SVD in Movielens 100K and 1M (see e.g. http://surpriselib.com/) therefore we expect it to run much slower than Fast Greedy as well. We think that a competitive implementation and performance benchmarking of Fast Greedy in the context of matrix completion will undertake some effort but will be important future work.
>
> **(iii)** We have not attempted to give explicit bounds on the condition number, and there are mainly two reasons for this: It is a quantity that is hard to compute, and in some practical instances this parameter might not even be bounded. In particular, for the setting of Figure 1 and 3, the restricted condition number is infinity, because the Frobenius norm only acts on a subset of elements. Still, the theoretical performance as a function of the restricted condition number has strong predictive power on the practical performance. These observations have also been made in the simpler case of sparse linear regression, where even though real instances can trivially have an infinite condition number (e.g. if a single feature vector appears twice), good restricted condition number-based bounds are still indicative of good practical performance. An interesting read for rank-restricted condition number in practice is Neghaban, Ravikumar, Wainwright, and Yu, 2012 (https://projecteuclid.org/download/pdfview_1/euclid.ss/1356098555) which proves that, in various settings, the "effective" rank-restricted condition number can be bounded with high probability, even though it is infinity in the worst case.

---

> > ### Author Response · Authors · 2020-11-24
> > **Author response (2/2)**
> >
> > **(iv)** We have added the following paragraphs in order give some intuition about the algorithms and techniques used.
> >
> > "Algorithm 1 works by iteratively adding a rank-1 matrix to the current solution. This matrix is chosen as the rank-1 matrix that best approximates the gradient, i.e. the pair of singular vectors corresponding to the maximum singular value of the gradient. In each iteration, an additional procedure is run to optimize the combination of previously chosen singular vectors.
> >
> > One drawback of Algorithm 1 is that it increases the rank in each iteration. Algorithm 2 is a modification of Algorithm 1, in which the rank is truncated in each iteration. The advantage of Algorithm 2 compared to Algorithm 1 is that it is able to make progress without increasing the rank of $A$, while Algorithm 1 necessarily increases the rank in each iteration. More specifically, because of the greedy nature of Algorithm 1, some rank-1 components that have been added to A might become obsolete or have reduced benefit after a number of iterations. Algorithm 2 is able to identify such candidates and remove them, thus allowing it to continue making progress."
> >
> > "The main bottleneck in Shalev-Shwartz et al. (2011) is the fact that the analysis is done in terms of the squared nuclear norm of the optimal solution. As the worst-case discrepancy between the squared nuclear norm and the rank is $R(0)/\epsilon$, their bounds inherit this factor. Our analysis works directly with the rank, in the spirit of sparse optimization results (e.g. Shalev-Shwartz et al. (2010), Jain et al. (2014), and Axiotis & Sviridenko (2020)). A challenge compared to these works is the need for a suitable notion of "intersection" between two sets of vectors. The main technical contribution of this work is to show that the orthogonal projection of one set of vectors into the span of the other is such a notion, and, based on this, to define a decomposition of the optimal solution that is used in the analysis."

---

### Author Response · Authors · 2020-11-25
**Revised manuscript**

We thank all the reviewers for the valuable feedback. We have uploaded an updated manuscript that incorporates the reviewers' comments. We hope that these changes address the comments raised by the reviewers.

---

### Decision · Program_Chairs · 2021-01-07
**Final Decision**

**Decision:**

Accept (Poster)

**Comment:**

Please clarify as early as the abstract that you refine the analysis of the algorithm proposed by Shalev-Shwartz et al (which is a great contribution given the importance of the problem).